# Influence of Elevated Temperatures and Cooling Method on the Microstructure Development and Phase Evolution of Alkali-Activated Slag

**DOI:** 10.3390/ma15062022

**Published:** 2022-03-09

**Authors:** Hua Fu, Rui Mo, Penggang Wang, Yanru Wang, Yubin Cao, Wentao Guang, Yao Ding

**Affiliations:** 1School of Civil Engineering, Qingdao University of Technology, Qingdao 266033, China; fs215379@163.com (H.F.); morui122@163.com (R.M.); guangwt1991@163.com (W.G.); dingyao0807@163.com (Y.D.); 2Centre for Future Materials, University of Southern Queensland, Toowoomba, QLD 4350, Australia; yubin.cao@usq.edu.au

**Keywords:** alkali-activated slag, elevated temperature, cooling method, durability, phase evolution

## Abstract

The performance of alkali-activated slag (AAS) under thermal treatment has received particular attention. In this study, the effect of five elevated temperatures (25, 200, 400, 600, and 800 °C) and two cooling methods (air cooling and water spraying) on the mechanical and durability properties, microstructure, and phase evolution of AAS was investigated. The results show that AAS mortars exhibit higher resistance to thermal attack than OPC in terms of strength and durability. AAS samples cooled in air show higher residual strength than those cooled by spraying water, which is mainly attributed to fewer cracks formed in the former. The resistance to carbonization of exposed AAS mortars depends on the pore size distribution, while that to chloride ion penetration depends on the porosity. Cooling methods show a minor effect on the phase evolution of reaction products, suggesting that the microstructure degradation is mainly responsible for the damage of AAS structures. This study provides fundamental knowledge for the thermally induced changes on AAS which contributes new ideas for the development of construction structures with higher fire resistance.

## 1. Introduction

The consumption of ordinary Portland cement (OPC), which is the basic material for construction buildings, has remained high in recent decades due to the continuous acceleration of global urbanization [1]. There are around 0.87 tons of greenhouse gas emissions during the production of 1 ton of OPC [2]. The OPC products are generally perceived to be fire-resistant, but with a significant deterioration of the mechanical strength and spalling at high temperatures above 400 °C [3]. The rapid loss of free water and bound water from the hydration products (calcium silicate hydrate and calcium hydroxide) of OPC is regarded as responsible for the strength loss. The spalling of the protection layer of reinforcement products could expose the steel bar to the fire environment and cause the building to collapse. Thus, alkali-activated materials (also called geopolymers) have received attention as their low carbon footprint and higher resistance to elevated temperature compared to OPC [4,5,6,7]. The geopolymer derived from alkali-activated slag (AAS) can be applied as construction material in specific applications that need a higher resistance to fire or high temperature [8,9,10,11]. In addition, due to the wide variety of building materials, the chemical components are diverse. In the case of fire, the indoor or air temperature can reach different levels, with values even as high as 800 °C. Therefore, four temperatures of 200 °C, 400 °C, 600 °C, and 800 °C were taken as the research object.

Although recent studies showed the high residual strength and the advantages of AAS in high-temperature applications [11], the explanations regarding the reason for the degradation of the strength after high temperatures are inconsistent. AAS shows higher residual flexural strength and lower mass loss when exposed to high temperatures from 200 to 800 °C and cooled in air, which is due to the increased thermal incompatibility between matrix contraction and aggregate expansion in AAS concrete [12]. The great thermal shrinkage and increased micro cracking in both AAS and OPC contribute to the strength loss [3]. However, the dehydration of calcium hydroxide (CH) is thought to be the main reason for the thermal shrinkage of OPC paste, while less or no formation of CH in AAS is a reason for its higher fire resistance [13]. AAS cured at room temperature shows slightly increased compressive strength when exposed to 200 °C, which is due to the further hydration of unreacted raw materials [14].

The cooling method is an influential factor that cannot be ignored when investigating the performance of AAS after exposure to fire. Two cooling methods are proposed considering practical application, i.e., air cooling and water spraying. Most studies regarding thermal attack focused on air cooling [15], while rare studies compared the difference between air cooling and water spraying. Water spraying may cause rapid shrinkage and cracks on the surface of the structure, but the water may contribute to the rehydration of the exposed materials, making up for the loss of calcination. The effect of spraying water on AAS and OPC exposed to different temperatures is not clear.

The service life of a construction building exposed to high temperature is another concern when studying the fire resistance of building materials, as the effect of the thermal attack on durability is still not understood. The durability of construction structures highly depends on the pore structure [16,17]. The low porosity of AAS incorporated with fine silica fume shows higher resistance to chloride ion penetration [18]. Similarly, reduced porosity and average pore size in AAS are reported as the main reasons for its higher resistance to carbonization [19,20]. The carbonization of AAS is also related to the reaction products formed during hydration and the alkalinity of the pore solution, which are also influenced by the thermal treatment and cooling methods. Thus, the influence of the thermal attack and cooling methods on the resistance to carbonization and ion penetration should be explored.

To investigate the effect of thermal attack and cooling methods on the mechanical properties and durability of the AAS, this study evaluates the performance of AAS mortars under different evaluated temperatures, i.e., 25, 200, 400, 600, and 800 °C, and two cooling methods (air cooling and water spraying). To better understand the change in microstructure and phase evolution, technical characterizations of the AAS paste exposed to different evaluated temperatures and cooling methods were applied, such as low-field nuclear magnetic resonance (LF-NMR), X-ray diffraction, and Fourier-transform infrared spectroscopy (FTIR). This research provides an experimental and theoretical basis for AAS in special applications with high temperatures.

## 2. Experimental Program

### 2.1. Materials

Ground granulated blast furnace slag (GGBFS) and OPC (type 52.5) were collected from Qingdao Municipal Group Concrete Industry Engineering Co., Ltd., Qingdao, China. The chemical composition of GGBFS and OPC analyzed by X-ray fluorescence (XRF) that is made by German Brooke manufacturer is given in Table 1. GGBFS is classified as alkaline with an alkalinity coefficient of 1.18 and hydraulicity modulus of 2.05 as per GB/T 18046-2017 [21].

River sand with a fineness modulus of 2.4 was used as fine aggregate, which belongs to the category of middle sand [22]. The activator solution for activating the GGBFS was prepared by dissolving sodium hydroxide powder (99 wt.% purity) in sodium silicate solution (Na_2_O = 8.54% (mass), SiO_2_ = 27.3%) and water to reach a modulus (SiO_2_/Na_2_O) of 1.8 and alkali content (Na_2_O) of 9 wt.% of the precursor powders. The water-to-binder ratio of the activator was set to 0.61 to reach a similar compressive strength to the OPC samples. The activating solution was allowed to equilibrate at room temperature prior to use.

### 2.2. Specimen Preparation

The specimens were prepared under laboratory conditions, and the mixed proportions of the AAS and OPC are shown in Table 2. For AAS mortars, GGBFS and sand were mixed for 1.5 min with a mortar mixer which is made by Shanghai Nanfang Pavement Machinery Co., Ltd., Shanghai, China. The activator was then added, and the mixture was mixed for another 2 min. According to [23,24,25], the fresh mortar was cast in cubic molds (70.7 mm × 70.7 mm × 70.7 mm) for visual appearance and carbonization tests, rectangular molds (40 mm × 40 mm × 160 mm) for strength tests, and cylindrical molds (Φ 100 mm × 50 mm) for the chloride ion penetration test. The AAS pastes were prepared by mixing GGBFS and activators for 2 min and cast in the rectangle molds (40 mm × 40 mm × 160 mm). After 24 h of curing at room temperature, the specimens were demolded and transferred to the water bath for another 27 days of curing. The OPC mortars and pastes followed a similar preparation process. It should be noted that the water reducer was based on the weight percentage (%) of OPC and mixed with water first before being mixed with the dry materials. For the convenience of classification and description, the air cooling of alkali-activated mortar specimens and ordinary Portland cement mortar specimens with the same strength is represented by “A”, water spraying is represented by “W”, and the temperature is represented by the highest number. For example, specimens excited by water glass heated at 600 °C and cooled by air can be expressed as “AAS-A-600”.

### 2.3. Testing Procedures

#### 2.3.1. High-Temperature Test and Cooling Test

All samples were pre-dried to a constant weight in an oven at 105 °C before the high-temperature test to avoid bursting due to rapid evaporation of water. An electric furnace (SRJJ-20-13) with a voltage of 380 V and power of 10 kW was used, which is made by Hefei Fushe Thermal Equipment Co., Ltd., China. Samples were heated to 200, 400, 600, and 800 °C at a heating rate of 10 °C/min, and held at the designed temperature for 2 h. A minor amount of steam began to overflow out the furnace after 150 °C during the calcination process (designated temperature of 200 °C). For samples exposed to 400 °C or above, there was a large amount of steam that came out from the furnace when the temperature reached approximately 300 °C. It should be noted that the OPC cylinders cracked at around 390 °C, and OPC samples were not exposed to 600 °C and 800 °C considering the safety issue. The details are shown in the next section.

Two cooling methods (air cooling and water spraying) were adapted. The air-cooled samples were taken out from the furnace immediately after the high-temperature test and cooled to room temperature on a laboratory bench (23 ± 2 °C, Rh = 45% ± 5%) for 24 h. The water-sprayed samples were cooled by spraying tap water on the surface of samples for 10 min (simulated fire sprinkler cooling) and then cooled on a laboratory bench for 24 h. The cooled samples were subjected to further tests.

#### 2.3.2. Physical Properties Tests

The change in appearance of cubic samples was observed using a digital microscope which is made by Shenzhen Chensheng Optical Instrument Co., Ltd., China. Three rectangular samples from each mix were first subjected to a flexural strength test with a DYE-300 electro-hydraulic servo pressure testing machine which is made by Shanghai Songgdun Instrument Manufacturing Co., Ltd., China at a loading rate of 50 kN/s (three-point loading with a span length of 100 cm). The fractured samples were used to test the compressive strength at a loading rate of 2.4 kN/s, which was determined according to a strength grade greater than 5 MPa [24].

#### 2.3.3. Durability Test

The cooled mortars for the carbonization test and water absorption test from each mix were dried at 60 °C oven for 48 h before durability tests. The carbonization test was conducted on one surface of the cubic mortars, with the remaining surfaces waxed. The cubic mortars then were kept in a carbonization chamber (THB, Jinggangyuan) at a temperature of 20 ± 3 °C, relative humidity of 70% ± 5%, and CO_2_ concentration of 20% ± 3%. After 3 days, 7 days, 14 days, and 28 days, samples were taken out from the chamber and split in the middle. The phenolphthalein alcohol solution (1%) was sprayed on the fracture surface. The carbonization depth and calcination depth were tested with 10 test points per edge by a vernier caliper (as shown in Figure 1). The carbonization depth of each specimen could be calculated by subtracting the final calcination depth from the average of the carbonization and calcination depth.

A water absorption test was conducted according to the previous study [26] with cubic mortars. The absorbed water amount was recorded with intervals such as 15 min, 30 min, 1 h, 1.5 h, 2 h, 2.5 h, 3 h, 4 h, 5 h, 6 h, 7 h, 8 h, 24 h, and 48 h.

A chloride diffusion test by the rapid chloride migration (RCM) method was conducted with cylindrical mortars according to the general guidelines provided by NT Build 492 and GB/T 50082-2009, as well as a previous study [27]. The cylindrical mortars were saturated in a vacuum saturation instrument which is made by Cangzhou Yixuan Test Instrument Co., Ltd., China, and the surfaces that formed the sides of the cylinder were sealed by wax. Figure 2 shows the details regarding the testing process. It should be noted that carbonization and chloride ion penetration require pre-suctions of the dried specimens before the test.

#### 2.3.4. Characterization Test

X-ray diffraction (XRD) patterns of AAS and OPC pastes were collected using a high-temperature in situ XRD instrument in the scan range of 10°–50° 2θ with a slow scan rate. The samples were heated with a heating rate of 10 °C/min and held at the designed temperature for 5 min before collecting the patterns. The Fourier-infrared (FTIR) spectrum which is made by Tianjin Jingtuo Instrument Technology Co., Ltd., China was collected by a Renishaw 71J012. The samples were prepared with a 100–200 mg KBr pellet with around 1–2 mg of pastes. The absorption mode was adapted in the scanning range of 4000–400 cm^−1^. The low-field NMR (LF-NMR) spectrum which is made by Suzhou Niumai Analytical Instrument Co., Ltd., China was collected using a Niumag PQ001 spectrometer with a Carr–Purcell–Meiom–Gill (CPMG) sequence. The magnetic field frequency was 18 MHz, and the magnetic field intensity was 0.42 T.

## 3. Results and Discussion

### 3.1. Physical Properties Analysis

#### 3.1.1. Visual Appearance

Figure 3 shows the visual appearance of AAS and OPC mortars after being exposed to different elevated temperatures and cooling methods. As the exposure temperature increased, the cracks generated on the surface of the AAS mortars, cooled in air or by spraying water, gradually increases and widened. At 400 °C, AAS mortars cooled by water spraying demonstrated significant surface peeling. When the temperatures increased to 600 and 800 °C, wide cracks formed on the surface of AAS mortars. The samples cooled in air showed more small cracks than those cooled by spraying water. This might have been due to the rehydration on the surface of samples when sprayed by water. On the other hand, water is cooler than air. Additional pressure was generated due to uneven cooling during watering, which led to spalling of the surface layer of the specimen.

For OPC mortars exposed to 200 °C, the pastes and sand started to peel off from OPC mortars, especially for samples cooled by water spraying. When the temperature increased to 400 °C, connected cracks appeared on the surface of OPC samples. The cracks broadened from 0.033 mm on the surface of samples cooled in air to 0.064 mm when the samples were cooled by water. The above results are basically consistent with the research results of Shoaib [28].

The free water began to evaporate from the samples, along with the generations of shrinkage and cracks, when mortars were exposed to 200 and 400 °C. At higher temperatures, the reaction products underwent dehydration, causing significant shrinkage and cracking. AAS mortars showed higher resistance to shrinkage and cracking than OPC mortars, which suggests less emission of water. Although water spraying caused wide and large cracks on the surface of the sample, the rehydration of the sample surface could reduce the formation of microcracks. The above results are consistent with the research results of Colins [29].

#### 3.1.2. Compressive and Flexural Strength

Figure 4 shows the compressive strength of each group of specimens before and after exposure to different elevated temperatures and cooling methods. The compressive strength of AAS mortars cured at 25 °C for 28 days reached 85 MPa. The compressive strength of AAS mortars increased after exposure for 2 h at a 200 °C furnace, while the compressive strength of samples cooled in air increased to 108 MPa (by 27%) and that cooled by water spraying increased to 97 MPa (by 14%). High temperature promoted further hydration of the unreacted components in AAS samples at 200 °C along with the formation of additional calcium aluminosilicate hydrates (C-A-S-H) which are the source of compressive strength [30]. The inside of AAS mortars stayed at a reasonably high temperature (200–250 °C) and high pressure (>1.0 MPa), which promoted further hydration [31]. The compressive strength of AAS mortars gradually declined as the exposure temperature increased from 200 to 800 °C. Specifically, the compressive strength of AAS was reduced by 87.8% and 86.1% after exposure for 2 h at 800 °C. This was due to the decomposition of the C-A-S-H gels and the formation of connected cracks in the samples, further reducing the compressive strength of samples.

For OPC mortars, the samples cured at 25 °C showed around 82 MPa of compressive strength, comparable to the AAS mortars. The strength of samples exposed to 200 °C increased to around 88 MPa (increased by 7%) after cooling in air and decreased to 77 MPa (decreased by 6%) after water spraying. When the treatment temperature increased to 400 °C, the compressive strength of OPC mortars was slightly lower than that at 200 °C. As shown in Figure 3, the mortars exposed to 400 °C generated obvious cracks but minor strength loss. This indicates that the cracks formed on the surfaces of samples were not penetrating. The influence of cracks on the compressive strength was less than the rehydration effect. This is different to the AAS mortars that showed a continuous decline in compressive strength with the minor formation of cracks upon exposure to 400 °C. This might have been due to the dehydration and decomposition of C-A-S-H gel at 300–400 °C [32,33].

As known, samples cooled by water spraying promote the formation of additional calcium silicate hydrates (C-S-H) in OPC mortars, which help to increase the compressive strength. However, the formation of cracks due to a rapid temperature change causes a substantial loss of compressive strength.

In terms of the above results, AAS mortars showed a higher increase in compressive strength than the OPC sample when exposed to 200 °C. This was due to a larger amount of rehydration in AAS mortars than OPC mortars. There were different reaction products formed in AAS and OPC, with more C-A-S-H gels formed in the former than C-S-H in OPC when exposed to 200 °C. On the other hand, the difference in the coefficient of thermal expansion between C-A-S-H and fine aggregates is smaller than that between C-S-H and fine aggregates, confirmed by the lower formation of cracks in AAS samples. Although AAS samples showed a significant loss of compressive strength (dropped by 78%), the integrity of the sample was still maintained. The above results are basically consistent with the research results of Hubler [34].

Figure 5 shows the flexural strength of each group of specimens exposed to different elevated temperatures and cooling methods. The flexural strength of all groups of specimens decreased gradually with the increase in heating temperature. After being exposed to 200 °C for 2 h, the flexural strength of AAS mortars cooled in air and by water spraying decreased by around 34.0% and 47.1%, respectively, while that of the OPC mortars decreased by around 4.3% and 23.5%. The flexural strength of AAS mortars continually decreased by 90% as the temperature increased to 800 °C.

According to Gu et al. [35], the rapid reaction at elevated temperature led to large chemical shrinkage in the specimen, and it also led to the formation of microcracks in the early stage. The flexural strength of AAS mortars decreased more quickly than that of OPC mortars, suggesting that the flexural strength of AAS mortars is more sensitive to thermal damage than OPC mortars [35,36]. The flexural strength of AAS mortars was much lower than that of OPC mortars when exposed to 400 °C, with the latter showing more cracks on the surface. There are two reasons related to this phenomenon. The original flexural strength of OPC is higher than that of AAS, indicating the higher flexural strength of C-S-H gel than C-A-S-H gel. On the other hand, the negative impact caused by cracks on flexural strength is also less than the effect of hydration products. The effect of cooling methods on the flexural strength of AAS mortars was not as significant as on the compressive strength. Water spraying of the OPC mortar exposed to 200 °C had a significant negative effect on its flexural strength. At 400 °C, the significant decrease in flexural strength reduced the difference in cooling methods.

### 3.2. Durability Performance

#### 3.2.1. Carbonization

Figure 6 shows the carbonization depth of AAS and OPC mortars during exposure to the carbonization test. The carbonization depth increased with the exposure time. The carbonization depth of AAS mortars cured at 25 °C was around 7 mm after being exposed to the CO_2_ chamber for 3 days and increased to 18 mm after 28 days. The OPC mortars cured at 25 °C showed much lower carbonization depth (less than 3 mm after 28 days).

After 200 °C calcination, the carbonization depth of AAS mortars cooled in air and by water spraying decreased to 4 mm and 6 mm at 3 days, respectively. Then, the carbonization depth of AAS mortars increased rapidly to around 35 mm and 41 mm at 28 days. The carbonization depth of AAS mortars increased significantly upon an increase in temperature above 400 °C. The AAS mortars exposed to 600 °C and cooled in water reached 70 mm of carbonization depth at 3 days. For AAS mortars exposed to 800 °C, the carbonization depths of samples cooled in air and by water spraying reached around 70 mm after 3 days of exposure to the CO_2_ chamber. At 28 days, the carbonization depth of AAS mortars exposed to temperatures over 400 °C reached 70 mm regardless of the cooling method. The samples cooled by water spraying showed higher carbonization depth compared to those cooled in air, suggesting higher porosity of the latter, which is consistent with the strength properties. The OPC mortars showed obviously less carbonization depth than AAS mortars, related to the different carbonization mechanisms in AAS and OPC mortars. The CH inside OPC mortars firstly reacts with the CO_2_ and then C-S-H gels, while the C-A-S-H gels in AAS directly participate in the carbonization [37]. Thus, the loss of alkalinity in AAS was faster, and the carbonization depth determined by the color response was higher than in OPC mortars. In terms of the above results, water spraying significantly increased the carbonization depth of AAS and OPC mortars.

#### 3.2.2. Chloride Ion Penetration

Figure 7 shows the chloride (Cl^−^) diffusion coefficient (CDC) of AAS and OPC mortars under different cooling methods. The AAS mortars exposed to 800 °C were not suitable for the chloride diffusion test, as the significant cracking inside AAS samples posed a high risk of the electrode plates burning. Due to sample destruction, OPC mortars exposed to temperatures above 400 °C were also unsuitable for the ion penetration test. The unexposed AAS and OPC mortars showed similar initial chloride diffusion coefficients, indicating their similar pore structures before exposure to thermal attack. After being exposed to 200 °C, the CDC of OPC mortars increased significantly, especially when cooled by water spraying, which displayed a 47% higher value. This was mainly related to the change in pore structures in the samples. Thermal attack caused the formation of connected cracks inside samples, promoting the penetration of Cl^−^ ions. When the exposure temperature increased to 200 °C, the CDC of AAS mortars reduced slightly, before increasing dramatically with temperature. The reduced CDC values confirm the continued hydration and promoted pore structure when AAS samples were exposed to 200 °C. Upon an increase in temperature, the formed cracks inside samples accelerated the penetration of Cl^−^ ions. In terms of the above results, the thermal attack significantly reduced the resistance of AAS and OPC samples to chloride ion penetration, while cooling methods had a minor influence on the CDC of the AAS mortars. The change in durability of samples was strongly related to the change in pore structure. The above results are basically consistent with the research results of Zhu [16,38].

### 3.3. Pore Structure Analysis

#### 3.3.1. Low-Field NMR Analysis

The pore size distribution and porosity have an important influence on the mechanical and durability performance of mortars. According to the distribution of pore size, Wu [39] divided pores inside concrete into harmless pores (≤20 nm), less harmful pores (20–50 nm), harmful pores (50–200 nm), and very harmful pores (≥200 nm). According to the bout model [40], the pore size can be divided into gel pores (1–10 nm), transition pores (10–100 nm), capillary pores (100–1000 nm), and macropores (>1000 nm). According to the bout model, to reflect the influence of harmful pores, the pore diameters were divided into the following intervals: <10 nm, 10–50 nm, 50–100 nm, 100–1000 nm, and >1000 nm.

Figure 8 shows the changes in porosity of specimens exposed to different elevated temperatures and cooling methods. The porosity of AAS mortars cured at 25 °C was 22.6%, while the porosity of OPC mortars was 11.3%. After being exposed to 200 °C, the porosity of AAS mortars cooled in air and water reduced to 12.5% and 13.5%, respectively. The reduced porosity was due to the rehydration of raw materials in AAS mortars with the formation of reaction products (C-S-H gel and C-A-S-H gel) [41]. The porosity increased gradually due to the dehydration of reaction products as the temperature increased from 200 °C to 600 °C, but remained lower than that before exposure to thermal treatment. At 800 °C, the porosity of AAS mortars was the lowest, related to the ceramization of the reaction products under high temperatures [42]. OPC mortars showed a continuous increase in porosity with temperature, while samples cooled in air showed lower porosity than those cooled by water spraying. This was due to the rapid cooling with water spraying causing higher temperature stress and further increasing the porosity of samples [43].

Figure 9 shows the percentage pore size distribution of AAS and OPC samples cured at 25 °C for 28 days. The harmful pores in the AAS mortar accounted for 13.8%, while the harmful pores in OPC mortar accounted for 37.2%. According to Figure 8 and Figure 9, AAS mortars showed higher porosity with a lower number of harmful pores (>100 nm) than OPC mortar specimens. The higher number of gel pores (<10 nm) of AAS mortars indicated a higher formation of reaction products [44].

Figure 10 shows the percentage pore size distribution of AAS mortars and OPC mortars under different cooling methods. From Figure 10a, with the increase in temperature, the percentage of less harmful pores (<100 nm) first decreased and then increased. At 200 °C, the AAS-A mortar still maintained 35% gel pores. When the temperature was greater than 200 °C, the number of gel pores in the AAS-A mortar overwhelmingly decreased. However, the AAS-W mortar showed a higher number of gel pores than the AAS-A mortar. This suggests that rehydration of mortars occurred when samples were cooled by water spraying. From Figure 10b, with the increase in temperature, the percentage of harmful pores first increased and then decreased when the temperature reached 800 °C. The proportion of gel pores in OPC-A mortars developed steadily with the increase in temperature. However, the proportion of gel pores in OPC-W mortars increased to the highest value when exposed to 400 °C.

The temperature and cooling methods had a significant influence on the pore size distribution of the mortars. The water-sprayed mortars showed a higher percentage of gel pores, indicating the rehydration of samples during the cooling process. AAS mortars cooled in air were composed of a higher number of harmful pores than those cooled by water spraying. OPC mortars showed a higher number of harmful pores than AAS before exposure to high temperature, the opposite pore size distribution to AAS mortars. The above results are basically consistent with the research results of [45,46].

#### 3.3.2. Capillary Water Absorption

Figure 11 shows the water absorption of AAS and OPC mortars cured at 25 °C for 28 days. There was an obvious linear relationship between the water absorption mass per unit area and the square root of time. The water absorption mass per unit area of the AAS mortar was greater than that of the OPC mortar, confirming the higher porosity of AAS mortars than OPC mortars.

Figure 12 and Figure 13 show the water absorption of OPC and AAS mortars under different cooling methods after elevated temperature treatment. For OPC mortars, the absorbed water derived by capillary force increased with the increase in absorption time and temperature. Comparing the two cooling methods, the water absorption of the air-cooled specimen was lower than that of the water-sprayed specimen at the same temperature, confirming the lower porosity of air-cooled samples. This suggests that water spraying caused more damage to the interior of the specimens than air cooling. For AAS mortars, the absorption rate of samples increased with the increase in temperature. In particular, when exposed to 800 °C, samples reached the highest absorbed water amount in the first 30 min. The pore structures analyzed by LF-NMR confirmed the water absorption amount. The cracks formed during water spraying enhanced the absorbed water amount and accelerated the absorbance rate.

#### 3.3.3. Relative Dynamic Elastic Modulus

Figure 14 shows the ultrasonic pulse velocity of AAS and OPC mortars after elevated temperature treatment. The ultrasonic pulse velocity of mortars decreased significantly with the increase in temperature. The decrease in pulse velocity was initially due to the loss of water in specimens. With the increase in temperature, the formation of cracks inside samples reduced the pulse velocity [47]. This was due to cracks inside the samples reducing the ultrasonic pulse velocity. However, samples cooled in water showed a higher ultrasonic pulse velocity due to the absorbed water refilling the small cracks inside samples. The ultrasonic pulse velocity could reflect the damage caused by high temperature, exhibiting no direct relationship with the compressive strength. AAS mortars showed increased compressive strength but reduced ultrasonic pulse velocity when exposed to 200 °C. The further hydration of raw materials could help improve the compressive strength, albeit with the generation of internal cracks. The AAS mortars after the 800 °C treatment showed the lowest porosity with the lowest ultrasonic pulse velocity. This was due to the transformation of hydration products in the specimen from the amorphous phase to the crystalline phase, which filled the pores and reduced the porosity. However, the internal damage of mortar caused by thermal shrinkage hindered the wave propagation and reduced the ultrasonic pulse velocity. The surface cracking of the OPC mortar specimen was not obvious after exposure to the elevated temperature of 200 °C. The decreased ultrasonic pulse velocity within 200 °C was caused by the removal of free water in the specimen. When the temperature increased to 400 °C, the ultrasonic pulse velocity of OPC mortar cooled in water was higher than that cooled in air. This is because the surface and internal cracks of the specimen increased after elevated temperature heating at 400 °C, and the water entered the cracks after spraying, resulting in a slow decline in ultrasonic pulse velocity. The above results are basically consistent with the research results of Han [48].

### 3.4. Phase Evolution

#### 3.4.1. XRD Analysis

Figure 15 shows the in situ XRD patterns of AAS and OPC pastes as the temperature increased from 25 °C to 800 °C. The AAS samples showed an amorphous hump at around 2θ/29°, with diffraction peaks characterized as C-S-H and C-A-S-H gel [32]. There was no diffraction peak of calcium hydroxide (CH). With the increase in temperature, the hump intensity of AAS paste was reduced due to the decomposition of C-S-H and C-A-S-H gels. At 600 °C, the diffusion peak of C-S-H and C-A-S-H phases almost disappeared. After heating to 800 °C, the diffraction peak of C-S-H in AAS paste disappeared with the formation of akermanite and gehlenite. This confirmed that the compressive strength of AAS mortars was highly related to the existence of amorphous C-A-S-H gels.

As shown in Figure 15 the OPC sample cured at 25 °C for 28 days showed the formation of C-S-H, CH, C_2_S, and C_3_S characteristic peaks, indicating that the specimen was not completely hydrated. With the increase in temperature, the diffraction peak intensity of C-S-H gradually decreased, suggesting that the elevated temperature promoted the dehydration and decomposition of C-S-H into C_2_S and C_3_S [49]. After heating to 600 °C, the diffraction peaks for CH disappeared, while the diffraction peak intensity of CaO became stronger. During the curing of cement paste, the cement paste was carbonized to form CaCO_3_. Decarburization of CaCO_3_ in cement paste occurs above 650 °C [50], transforming into CaO and CO_2_; thus, the diffraction peak of CaO increased in the 800 °C diffraction pattern.

#### 3.4.2. FTIR Analysis

Figure 16 shows the infrared spectrum of AAS paste after exposure to different temperatures. For the samples cured at 25 °C, the main broad band at around 970 cm^−1^ was attributed to the asymmetric stretching vibration of Al-O and Si-O bonds from a single tetrahedral ZO_4_ (Z=Si, Al) [51]. This band was an overlap formed by the superposition of stretching modes of the crystalline phase and glass phase. Puertas and Torres-Carrasco [52] showed that the asymmetric stretching vibration of the Si-O bond in the SiO_4_ tetrahedron existed in the absorption band of 967–971 cm^−1^, which is a significant feature of C-A-S-H in the sample. The second most dense band at 450 cm^−1^ was attributed to the typical in-plane bending mode of Al-O and Si-O bonds [53]. The contribution of Al-O vibration to the observed characteristics was lower, because the aluminum-to-silicon ratio in the AAS sample was around 1–6 [19]. The terminal OH groups from free water and bound water were represented by a wide band centered at approximately 3200 cm^−1^ and a weak band centered at approximately 1640 cm^−1^, respectively. The small band at around 1450 cm^−1^ indicated the stretching vibration of C-O bonds in carbonates.

During elevated temperature treatment, the strength of the diffusion bands at 3440–3450 cm^−1^ and 1640 cm^−1^ decreased due to the evaporation of free water and dehydration of reaction products. However, the hydroxyl group band at around 3450 cm^−1^ remained in the OPC samples at 800 °C, which was caused by water absorption during sample preparation. After 600 °C treatment, the peak for carbonates in samples disappeared due to the decomposition. At 800 °C, the smooth Si-O bond band of OPC paste was divided into four small bands at around 1020, 982, 935, and 850 cm^−1^, indicating the increased order of the silicate network, which is consistent with the XRD results. The dense zone at 935 cm^−1^ represented the extension mode of nonbridging oxygen in disilicate, which exists in akermanite above 700 °C. The other peaks may have been related to the extension of bridging and nonbridging oxygen atoms in the ZO_4_ tetrahedron in the crystalline phase or to a second extension in the glass phase. The band at 850 cm^−1^ was contributed by the asymmetric stretching vibration of the Al-O bond, which also proved the existence of gehlenite AAS paste after being exposed to 800 °C. The symmetric stretching and bending deformation of SiO_4_ and MgO_4_ groups exhibited several small bands between 600 and 750 cm^−1^. The peak at 680 cm^−1^ indicated that aluminum in the system appeared with a coordination number of four. The strong band at 473 cm^−1^ was caused by the vibration of ring silicate.

There were minor differences among the spectra of the OPC samples cooled in air and water. This was due to the water cooling having a minor effect on the functional groups of the hydration products. This suggests that cooling methods have a significant influence on the microstructure and further mechanical properties, with a minor influence on the reaction products.

Figure 17 shows the FTIR spectrum of OPC paste after elevated temperature, in which the main absorption peaks are marked. The absorption peak at 980 cm^−1^ of OPC paste cured at 25 °C was contributed by Si-O stretching vibration in C-S-H. The small peak at around 875 cm^−1^ and broad band at around 1420 cm^−1^ were recognized as the asymmetric vibration of the C-O bond in carbonates (${\mathrm{CO}}_{3}^{2-}$). The broad band at 3640–3700 cm^−1^ was contributed by the O-H vibration in free water, while the sharp peak at 3650 cm^−1^ was caused by OH^−^ stretching vibration in crystal water (i.e., CH). With the increase in temperature from 25 °C to 400 °C, the absorption peak at 980 cm^−1^ gradually widened with the decreased intensity of water and carbonates. After 600 °C, one can note the formation of two peaks at 1520 and 1400 cm^−1^ and the increased intensity of the band at 875 cm^−1^. This was due to the decomposed reaction products being carbonized during thermal treatment. It should be noted that CH existed in the samples exposed to 600 and 800 °C, which is in contrast to the XRD analysis results. This is because the XRD test was conducted with an in situ calcination instrument in a closed environment. However, the FTIR test was conducted with the cooled samples, whereby the CaO from the decomposition of reaction products reacted with water from air and water to regenerate Ca(OH)_2_. It was similarly concluded from the results of OPC pastes that cooling methods had a minor effect on the reaction products.

## 4. Conclusions

In this paper, the effects of different cooling methods on the microstructure, mechanical properties, and durability performance of AAS mortars after high-temperature treatments were investigated. The main conclusions drawn were as follows:(1)During thermal treatment, OPC mortars showed less resistance to thermal attack with the formation of wide cracks on the surface when the temperature increased to 400 °C. Samples cooled by spraying water showed significant peeling and wider cracks than those cooled in air, while the latter were filled with tiny cracks.(2)The compressive strength of AAS mortars increased first and gradually declined as the exposure temperature increased from 25 to 800 °C, while the samples cooled in air showed higher residual strength than those cooled by spraying water. The rehydration of AAS samples with spraying water could not offset the effect of thermal damage on the strength, which was even worse than in samples cooled in air. The effect of cooling methods on the flexural strength of AAS mortars was not significant.(3)The unexposed AAS mortars showed higher porosity but fewer harmful pores (>100 nm) than OPC mortars. After being exposed to thermal treatment, OPC mortars showed higher porosity than AAS mortars but with a lower formation of harmful pores. AAS mortars cooled by spraying water showed fewer harmful pores and more macropores (>1000 nm) than those cooled in air.(4)AAS mortars cooled in air showed a slower carbonization rate and less chloride ion penetration than those cooled by spraying water. OPC mortars showed higher chloride ion penetration but much less carbonization than AAS mortars. The water absorption and dynamic elastic modulus tests proved that more serious internal damage occurred inside water-sprayed samples.(5)The AAS pastes cooled in air and by spraying water showed comparable results regarding the phase change, suggesting a minor effect of cooling methods on the phase evaluation of mortars. During thermal treatment, the phase transformation of AAS paste was more obvious than that of OPC paste with higher crystallization.

## Figures and Tables

**Figure 1 materials-15-02022-f001:**
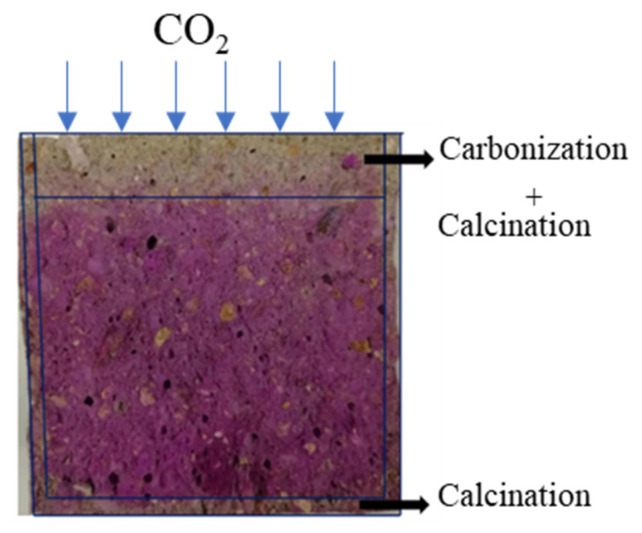
Cross-section image of the cubic mortar after phenolphthalein spray showing the depth of carbonization and calcination.

**Figure 2 materials-15-02022-f002:**
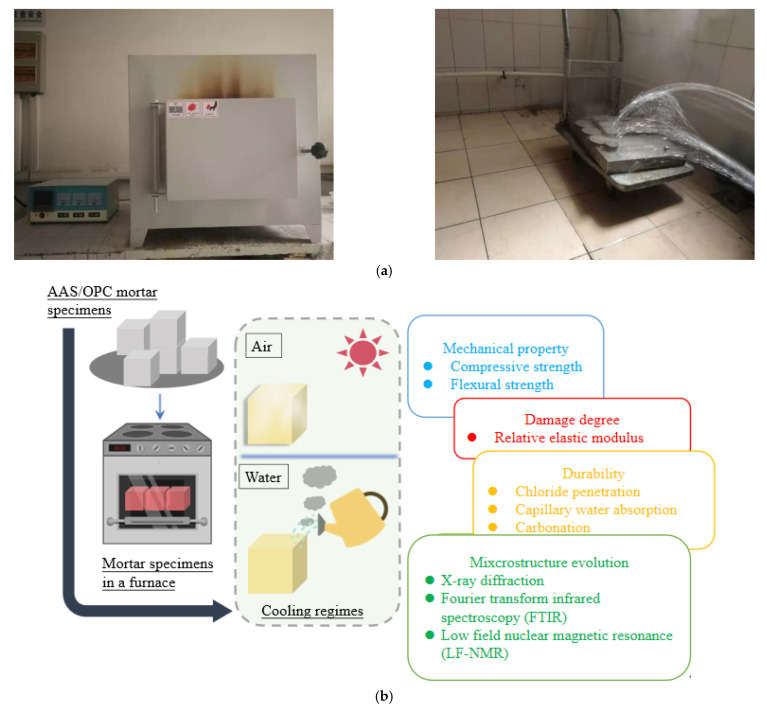
Testing process of this study: (**a**) images; (**b**) illustration.

**Figure 3 materials-15-02022-f003:**
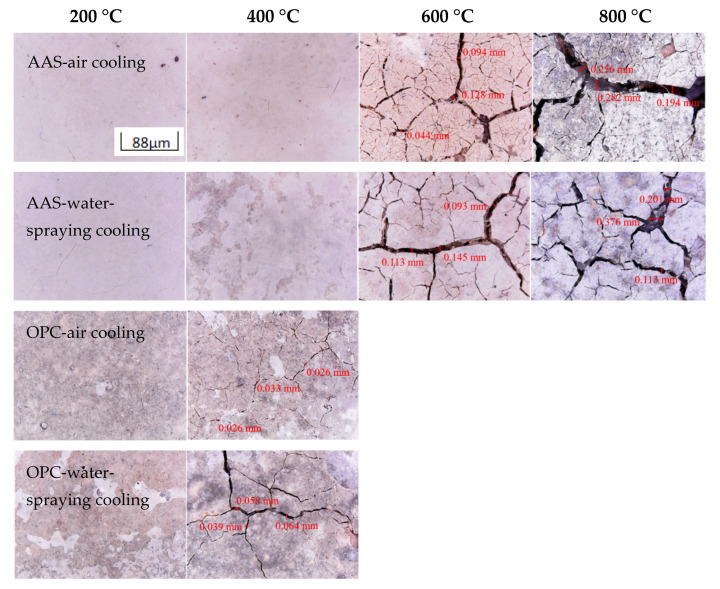
Photographs of AAS and OPC mortars after exposure to 200, 400, 600, and 800 °C thermal treatment and different cooling methods.

**Figure 4 materials-15-02022-f004:**
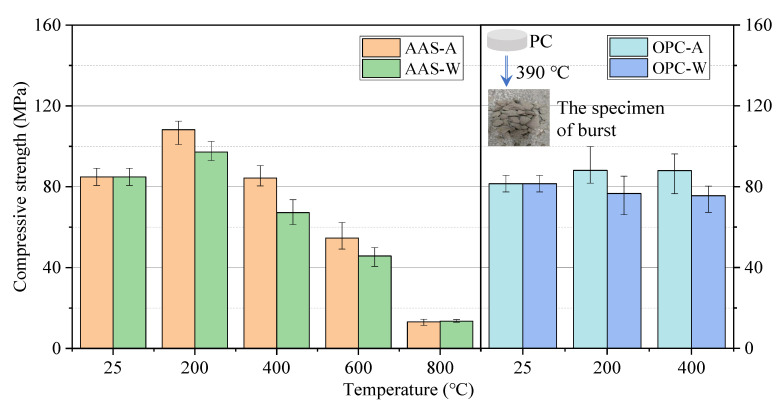
Compressive strength of specimens exposed to different elevated temperatures and cooling methods.

**Figure 5 materials-15-02022-f005:**
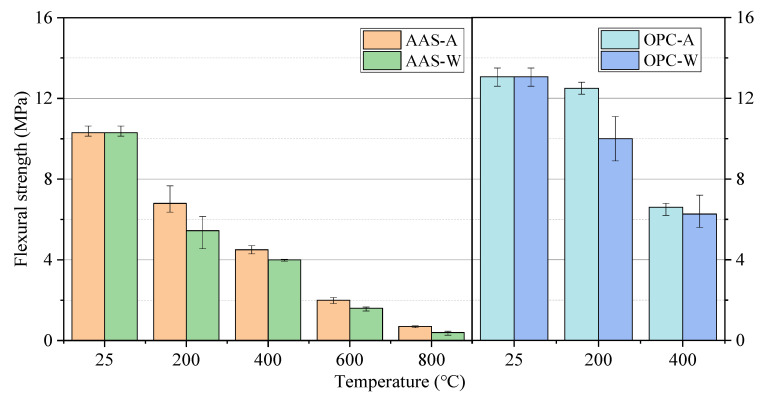
Flexural strength of mortar specimens exposed to different elevated temperatures and cooling methods.

**Figure 6 materials-15-02022-f006:**
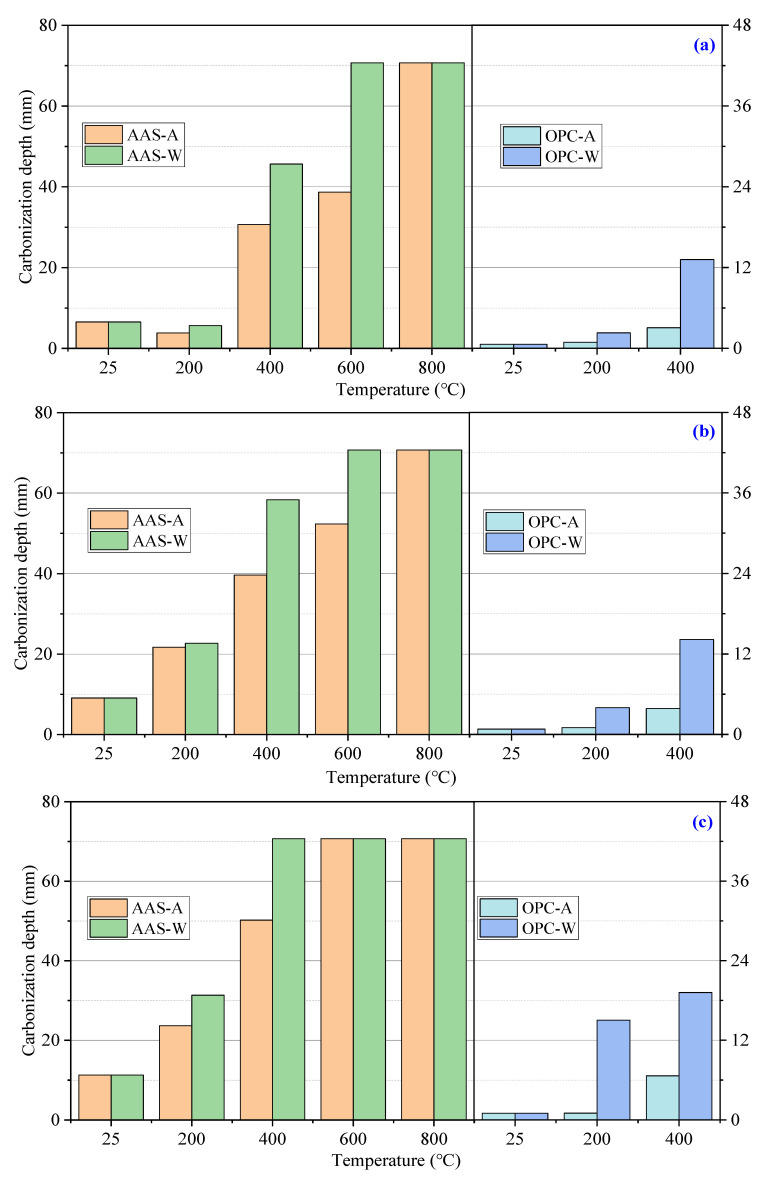
Carbonization depth of AAS and OPC mortars after carbonization for (**a**) 3 days, (**b**) 7 days, (**c**) 14 days, and (**d**) 28 days.

**Figure 7 materials-15-02022-f007:**
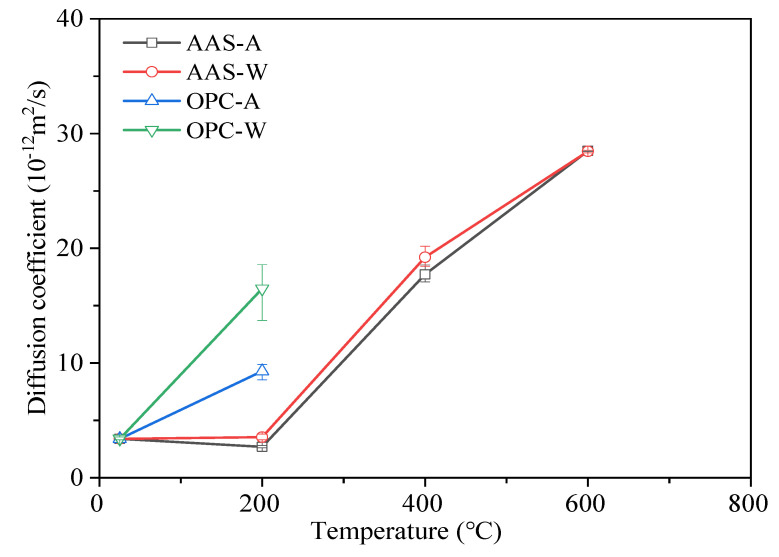
Chloride diffusion coefficients of specimens exposed to different elevated temperatures and cooling methods.

**Figure 8 materials-15-02022-f008:**
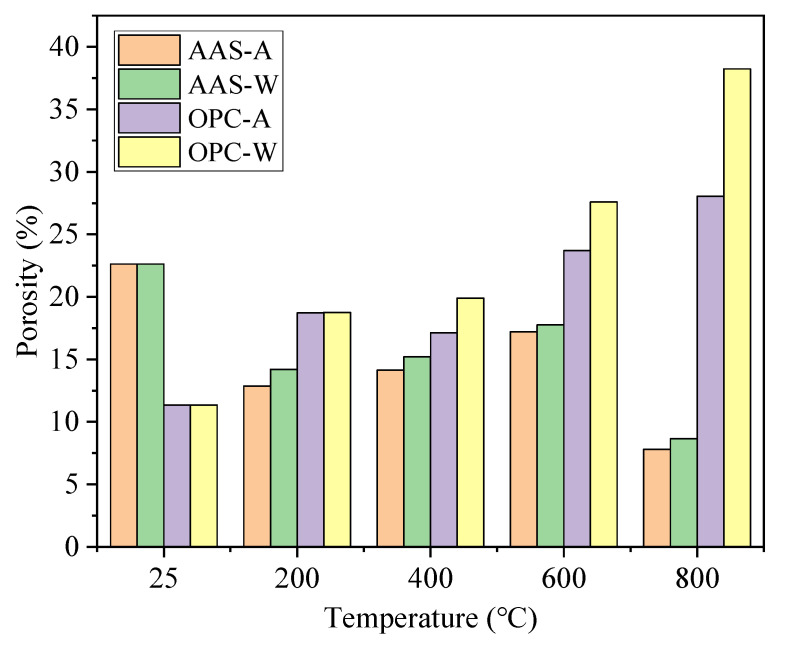
Changes in porosity of specimens exposed to different elevated temperatures and cooling methods.

**Figure 9 materials-15-02022-f009:**
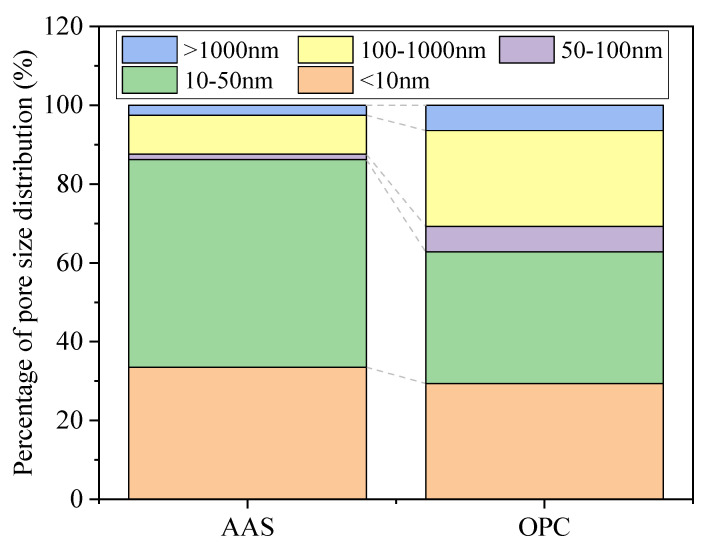
Percentage pore size distribution of specimens after 28 days of curing.

**Figure 10 materials-15-02022-f010:**
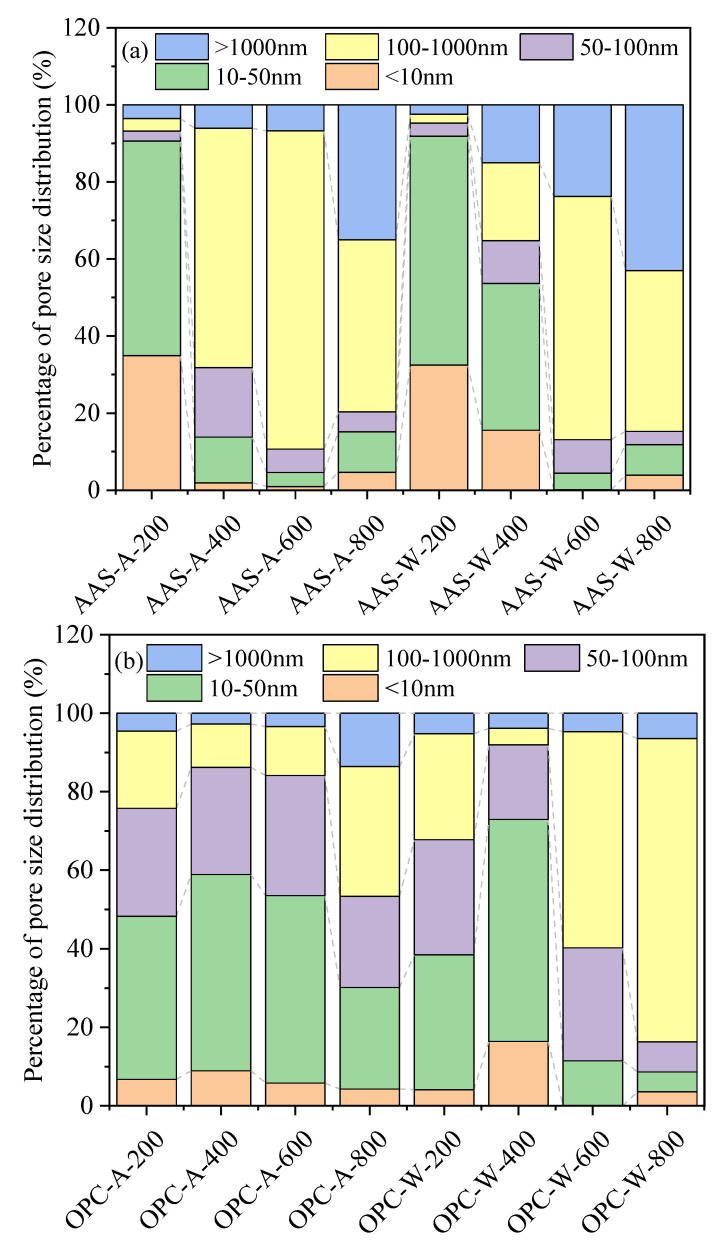
Percentage pore size distribution of specimens exposed to different elevated temperatures and cooling methods: (**a**) AAC specimens; (**b**) OPC specimens.

**Figure 11 materials-15-02022-f011:**
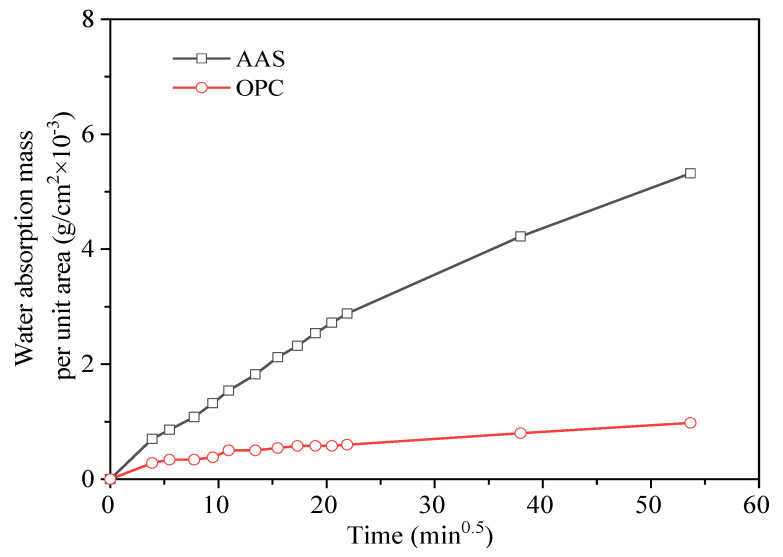
Relationship between water absorption mass per unit area and time of mortar specimen.

**Figure 12 materials-15-02022-f012:**
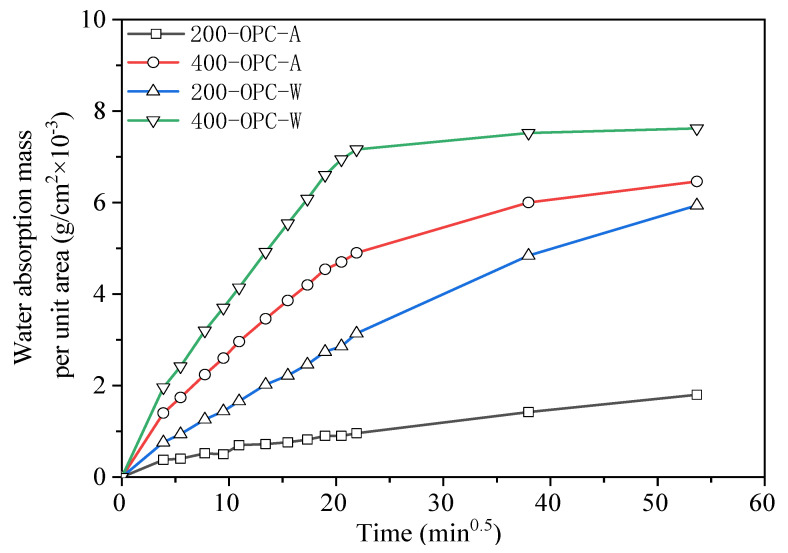
Relationship between water absorption mass per unit area and time of OPC mortar specimen under different cooling methods and elevated temperatures.

**Figure 13 materials-15-02022-f013:**
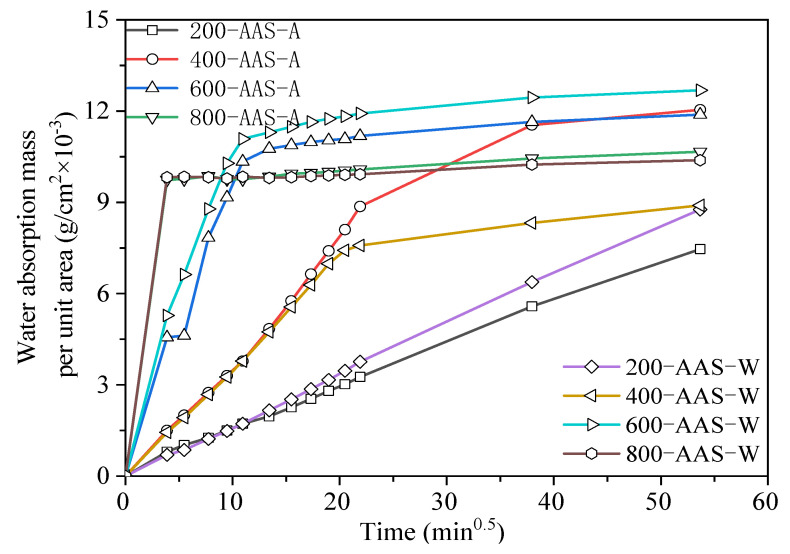
Relationship between water absorption mass per unit area and time of AAS mortar specimens exposed to different elevated temperatures and cooling methods.

**Figure 14 materials-15-02022-f014:**
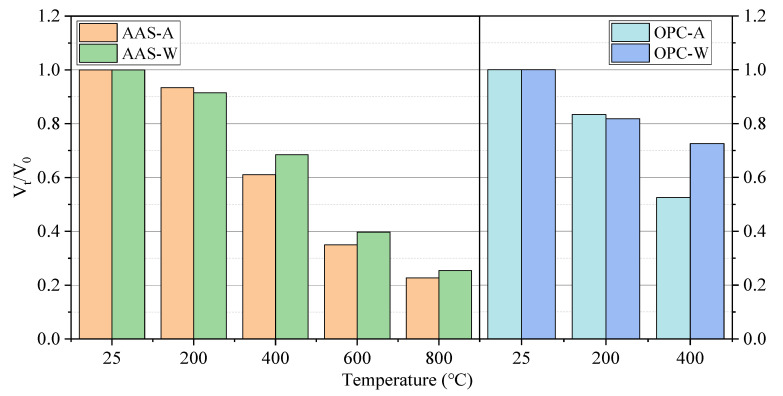
Relative ultrasonic pulse velocity of mortar specimens exposed to different elevated temperatures and cooling methods.

**Figure 15 materials-15-02022-f015:**
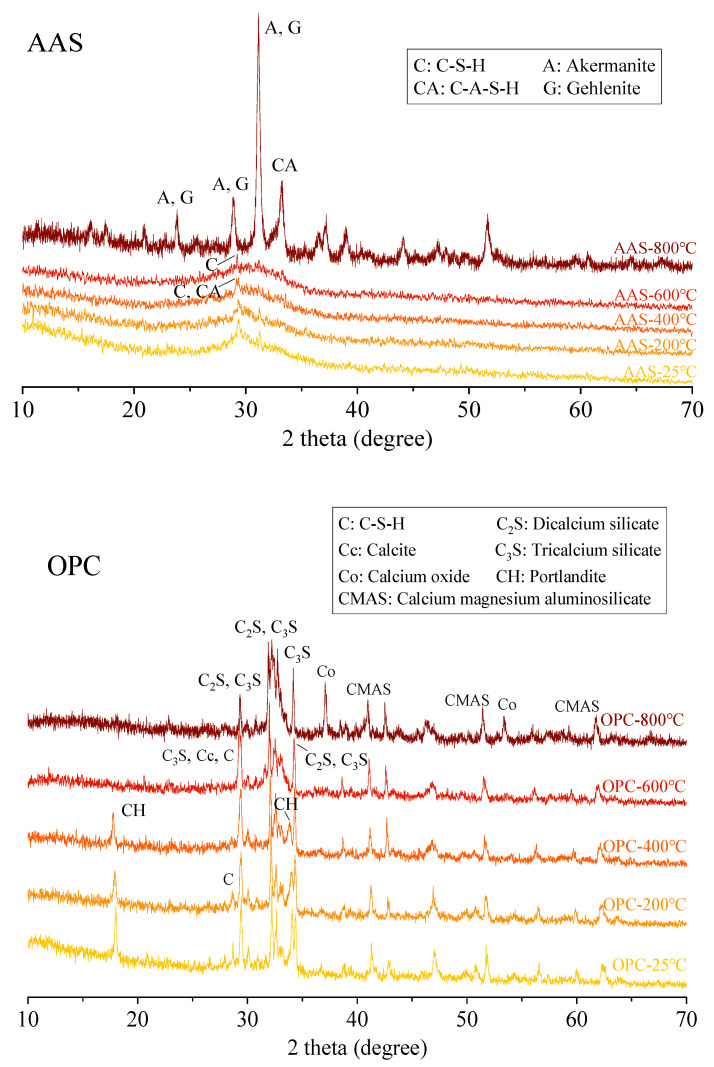
XRD results of cement paste after elevated temperature treatments.

**Figure 16 materials-15-02022-f016:**
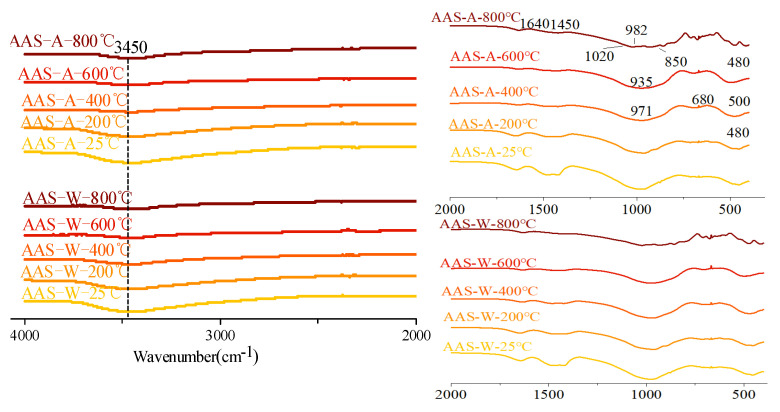
FTIR results of AAS paste exposed to different elevated temperatures and cooling methods.

**Figure 17 materials-15-02022-f017:**
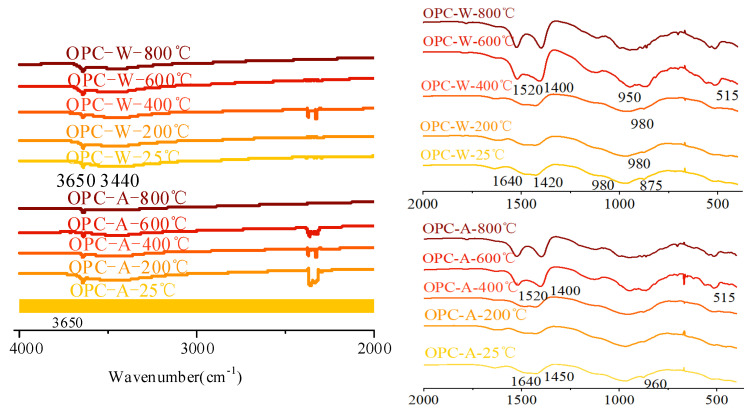
FTIR results of OPC paste exposed to different elevated temperatures and cooling methods.

**Table 1 materials-15-02022-t001:** Chemical composition of OPC and GGBFS.

Oxide (%)	CaO	SiO_2_	Al_2_O_3_	MgO	Fe_2_O_3_	SO_3_	K_2_O	Na_2_O	TiO_2_	MnO	SrO
OPC 52.5	58.62	21.44	4.82	6.71	3.16	2.75	0.89	0.37	-	-	-
GGBFS	41.32	31.34	12.52	10.27	0.45	1.93	0.36	-	0.88	0.22	0.1

**Table 2 materials-15-02022-t002:** Mix proportion of AAS and OPC specimens (kg/m^3^).

Mix Composition	Slag	OPC	Sand	Activator (g)	Water	Water Reducer (%)
AAS mortar	600	-	1200	1090.5	276	-
AAS paste	600	-	-	1090.5	276	-
OPC mortar	-	678	1650	-	122	3.5
OPC paste	-	678	-	-	122	3.5

## Data Availability

The data presented in this study are available on request from the corresponding author.

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
