# Peer review of "Influence of Elevated Temperatures and Cooling Method on the Microstructure Development and Phase Evolution of Alkali-Activated Slag"

_materials, 2022, doi:10.3390/ma15062022_

Round 1
Reviewer 1 Report
The authors have studied the “Influence of elevated temperatures and cooling regimes on the microstructure development and phase evolution of sodium silicate-activated slag”. The author used five high temperatures and two cooling regimes to explore the mechanical and durability properties, microstructures, and phase evolution of alkali-activated slag. This study may be potential and interesting to researchers. However, the current form of the manuscript is not suitable for publications and the authors should correct and clarify the following comment to further improve the quality of their work.
Line no: 213
The author reported that the OPC cement mortar compressive strength is around 82 MPa when cured at 25oC. Is it possible to attain this compressive strength in cement mortar without coarse aggregates and admixtures?
Line no: 214
The author reported that after heat treatment at 400oC, OPC cement mortar compressive strength shows similar at 200oC.
The authors should clarify the statement. For example, in Fig 3, the photographs of OPC cement mortars cured at 400oC show cracks on the surface with air cooling and water cooling, when compared to the sample cured at 200oC. After cracking’s are formed, how the compressive strengths are similar in both temperature and Cooling regimes?
At the same time, the AAS cement mortar cured at 400oC shows no cracks formation on the surface, but compressive strength found for the sample (air cooling and water cooling) is very less than OPC mortar cured at 400oC. How this happen? Is there any particular reason for that?
The compressive strength of AAS shows larger than cement mortar, when exposed to 200oC. Is it due to confirming its higher resistance to thermal attack? This is due to different reaction products formed in AAS and OPC, and the formation of the C-A-S-H gel in the former is higher which is resistant to thermal attack than C-S-H in OPC. However, the flexural strength will have reduced at the same temperature, -How? - And the same mechanism is not followed on the flexural strength specimens? why
At the same time (line no 239), the flexural strength will have reduced, the author told that it is due to the formation of microcracks in the early stage. The flexural strength of AAS mortars decreases faster than that of OPC mortars, suggesting the flexural strength of AAS mortars is more sensitive to thermal damage than OPC mortars [26, 27].
According to the Fig. 3, at the temperature of 400oC, the more number of surface cracks and microcracks are formed on the OPC mortar than AAS. As per the author's suggestion flexural strength is similar or little increased than OPC mortar.
But the flexural strength reduced, the author should clarify it.
The author should give a correlation of XRD and FTIR results with compressive strength.
Author Response
Your comments and suggestions have been uploaded in document. Please see the document.

Reviewer 2 Report
The presented work shows high-quality results that are of interest to be available to the interested readers and researchers. To improve the quality of the scientific writing, please consider introducing the following:
- Bear in mind that the alkali-activation is done by sodium hydroxide while adding sodium silicate participates in the construction of geopolymers. This word (geopolymer) was not used but should be mentioned in the paper. Geopolymer network is amorphous, and that is the other word that was rarely used in the text, although it is very important in these processes. Please, refer to the literature (10.1016/j.clay.2022.106410) while commenting on the changes in the quantity of amorphous or semi-crystalline matter in the materials obtained in XRD. How is that so that there is no peak corresponding to quartz in the mixture? At the expense of which minerals were geopolymers formed?
- The title should be improved since sodium silicate is not the activator. Besides, I suggest changing the expression “cooling regime” since it is not a regime but a cooling method or similar. You can use “cooling by air or water”. If the simulation of extinguishing the fire in a building is intended, that can be used in the title to draw attention.
- All of the methods applied and material combination, particle sizes used, curing and firing regime should be referenced if taken from the literature, or the choices in these terms must be explained. Besides, all the results obtained can be compared to the available literature for similar kinds of materials, or at least to OPC mortars.
- Are the FT-IR diagrams that are originally obtained collected along the Y-axis? The peaks look small. The peaks obtained should be presented in the real size to be visible and analyzed in terms of shifting the largest peak at around 1000 cm-1 by the geopolymerization process and changing the crystallinity of the system by using the crystallinity index of quartz (10.1016/j.clay.2022.106410). This should prove if the intended activation appears.
- Maybe there is another explanation for peeling of the AAS and OPC samples after the water-spraying process? Was the water cooler then the air, were there some extra stresses because of the uneven cooling during the initial watering?
- What temperatures are in the case of a fire in a building? Which temperatures should the mortar be able to handle? They could be higher than 1000 ⁰C. In these cases, how would you try to further improve the fire resistance and cooling by water or air, of the material you presented?
- Other minor issues are as follows: - Unbold “The” at the beginning of the Abstract. - The mixes in Table 2 could be parallel present in %. - Please, try to explain why the cylinders broke after firing at 400 ⁰C. - Explain the RCM abbreviation or use the full term since this appears only once in the text. - Which chemical is used for saturation in simulating Cl- attack? - Please change the furnace presented in Fig. 2, since that is certainly not the furnace used in the laboratory. It appears that there is some additional text in Fig. 2 which has been folded. Do not use 3 dots, replace it with text or delete it. - Change “high” to “higher” in line 191. - Explain the abbreviations used in a legend or below every image. What is meant by the “specimen of burst” in Fig. 4? - Figs. 9 and 10 are better to be presented as a table or comparable bars.
- The numbers in the references list are doubled.
Author Response
Your comments and suggestions have been uploaded in document. Please see the document.
Date: Feb. 23th, 2022
Dear Editor,
We are very grateful to you that give us an opportunity to revise and resubmit our manuscript. We also thank the reviewers for carefully reviewing the manuscript and providing extremely helpful feedback. We have made great efforts to revise the manuscript.
Reviewer’s comments are numbered and appear verbatim in blue words, authors responses appear immediately following each comment in black words.
Notice that all the numbers of lines mentioned below aim at the new submitting unless special statement. After completing revision, we submit two versions of the manuscript, i.e., one is the original version with revision mark (manuscript marked), where revision parts are marked in yellow words; another is the ultimate version (manuscript ultimate).
Reviewers 2 ' comments
The presented work shows high-quality results that are of interest to be available to the interested readers and researchers. To improve the quality of the scientific writing, please consider introducing the following:
- Bear in mind that the alkali-activation is done by sodium hydroxide while adding sodium silicate participates in the construction of geopolymers. This word (geopolymer) was not used but should be mentioned in the paper. Geopolymer network is amorphous, and that is the other word that was rarely used in the text, although it is very important in these processes.
The author modified the statement in the Introduction section. As shown in line 36-38. “Thus, alkali-activated materials (as called as geopolymers) have received more attention as their low-carbon footprint and higher resistance to elevated temperature compared to OPC [4-7]. The geopolymer derived from alkali-activated slag (AAS) is…”
- Please, refer to the literature (10.1016/j.clay.2022.106410) while commenting on the changes in the quantity of amorphous or semi-crystalline matter in the materials obtained in XRD. How is that so that there is no peak corresponding to quartz in the mixture? At the expense of which minerals were geopolymers formed?
The absence of quartz peak in the mixture means that the silicon is present in amorphous form. The ground granulated blast furnace slag used in this study composes around 31.3% of SiO2, much less than the clay. The clay usually sourced from minerals which contains higher content of quartz.
3.The title should be improved since sodium silicate is not the activator. Besides, I suggest changing the expression “cooling regime” since it is not a regime but a cooling method or similar. You can use “cooling by air or water”. If the simulation of extinguishing the fire in a building is intended, that can be used in the title to draw attention.
According to your suggestion, we have changed the title and article chapters.
4.All of the methods applied and material combination, particle sizes used, curing and firing regime should be referenced if taken from the literature, or the choices in these terms must be explained. Besides, all the results obtained can be compared to the available literature for similar kinds of materials, or at least to OPC mortars.
All of the methods applied and material combination, ect. are based on national codes and standards. And the main parameter values that can reflect the results have been proposed in the ordinary paper. Please verify it carefully.
All the results obtained have been compared to the available literature for similar kinds of materials. Please see the lines 211, 219, 261, 341, etc. In addition, some results have been compared to the available literature in the ordinary manuscript, for example, the lines 277, 312, etc.
5.Are the FT-IR diagrams that are originally obtained collected along the Y-axis? The peaks look small. The peaks obtained should be presented in the real size to be visible and analyzed in terms of shifting the largest peak at around 1000 cm-1 by the geopolymerization process and changing the crystallinity of the system by using the crystallinity index of quartz (10.1016/j.clay.2022.106410). This should prove if the intended activation appears.
The FTIR lines are originally obtained collected along the Y-axis. But the lines are proportionally scale down. The authors have modified the figures according to your suggestion.
6.Maybe there is another explanation for peeling of the AAS and OPC samples after the water-spraying process? Was the water cooler then the air, were there some extra stresses because of the uneven cooling during the initial watering?
Thanks for your suggestion. The authors have modified the statement in the revised manuscript, as shown in lines 204-206. “On the other hand, the water is cooler than air. Additional pressure is generated due to uneven cooling during watering, which leads to spalling of the surface layer of the specimen.”
7.What temperatures are in the case of a fire in a building? Which temperatures should the mortar be able to handle? They could be higher than 1000 ⁰C. In these cases, how would you try to further improve the fire resistance and cooling by water or air, of the material you presented?
In case of fire, the temperature may reach 1000 ⁰C, and we also considered this temperature when designing the experiment. Relevant tests were carried out, but 1000 ⁰C did great damage to the experimental instrument during the experiment, so we abandoned the effect of 1000⁰C on mortar. In addition, according to the relevant research, the buildings in reality have been damaged before the fire temperature reaches 1000⁰C due to the influence of load and other factors. And the effect of 1000⁰C is similar to the effect of 800⁰C on mortar properties.
8.Other minor issues are as follows: Unbold “The” at the beginning of the Abstract; The mixes in Table 2 could be parallel present in %.
The author has modified the small mistakes. The mixtures in table 2 refers to the amount of cementitious material, and its unit is kg/m3.
- Please, try to explain why the cylinders broke after firing at 400 ⁰C.
The reason for the bursting of the specimen at 400 °C may related to the lower porosity of the cement paste. Compared with alkali-activated slag, this will cause that the internal moisture of the specimen is unable to be eliminated, and then increase the pressure on the internal pores.
- Explain the RCM abbreviation or use the full term since this appears only once in the text.
RCM refers to the Rapid Chloride Migration test method. We have added it, as shown in line 176.
-Which chemical is used for saturation in simulating Cl- attack?
Deionized distilled water was used for sample saturation in simulating Cl- attack. Sodium chloride (10%) was used for the Cl- attack. More details regarding to the RCM test could refers to our previous study [10].
Please change the furnace presented in Fig. 2, since that is certainly not the furnace used in the laboratory. It appears that there is some additional text in Fig. 2 which has been folded. Do not use 3 dots, replace it with text or delete it.
The author modified the Fig. 2 according to your suggestion.
- Change “high” to “higher” in line 191.
The author changed the word.
- Explain the abbreviations used in a legend or below every image. What is meant by the “specimen of burst” in Fig. 4?
The author explained it, please see the lines 118-123. “For the convenience of classification and description, the air cooling of alkali activated mortar specimens and ordinary Portland cement mortar specimens with the same strength is represented by "A", the water spray cooling is represented by "W", and the temperature is represented by the highest number. For example, the specimens excited by water glass are heated at 600℃ and the cooling method is air cooling, which can be expressed as "AAS-A-600". The specimen of burst in Figure 4 refers to specimen broken.”
- Figs. 9 and 10 are better to be presented as a table or comparable bars.
Because the table and comparable bars are not conducive to observe the law of pore size distribution, we think it is appropriate to use the column.
-The numbers in the references list are doubled.
The numbers in the references list have been changed.

Reviewer 3 Report
It is a very interesting and valuable work written professionally.
I only have some minor comments:
Please add where the building/concrete is exposed to temperatures as high as 200 ° C and more, and information where these tests could be applicable.
Elastic modulus is a material property. Damage degree can be e.g. crack length.
Figure 3 - Add the bar scale.
line 212: Was measured the stiffness? How was it measured? What definition of stiffness was used?
Author Response

(The authors gave the same response as above.)

Reviewer 4 Report
The Authors studied the effect of thermal attack and cooling regimes on the mechanical properties and durability of the alkali-activated slag. The paper potentially contributes to the literature as it presents novel experimental results of interest for both research and practice purposes. However, the manuscript is affected by minor issues, and after minor revisions, it could be accepted for publication.
General report and comments:
- Line 8. Doubled word ‘Correspondence’. Please correct.
- Line 26, 172, 496. Lack of chapter number, see the Materials Template format.
- Line 89. Please add information on the grain-size distribution of the river sand.
- Line 101-104. The dimension of prepared specimens are nonstandard? Please add a reference to standards or codes to assumed dimensions of the prepared cubic, rectangular and cylindrical specimens.
- Line 122. The Authors gave reference to Fig. 3. The reference to Fig. 1 (line 145) and Fig. 2 should be given before Fig. 3. Please verify and correct.
- Chapter 2.3.1. Please add the photo of the laboratory test stand.
- Line 134-135. The assumed loading rate (2.4 kN/s) for compressive strength is very height. Please add a reference to code or standard where this value of loading rate is specified. Please give a comparison with other standards requirements.
- Line 233-37. Please verify with the Materials Template format.
- Results and discussion chapter. Please correct the sentences with the word ‘Fig.’ as the beginning of sentences, see Line: 145, 198, 233, 252, 282, etc.
- Conclusion chapter. Please, summarize the conclusions using bullet points. It would certainly emphasize the significance of the outcomes. Additionally, there should be closing remarks after the general conclusions (after the bullet points of conclusions), keeping in mind all the outcomes obtained.
Author Response

(The authors gave the same response as above.)

Round 2
Reviewer 1 Report
The manuscript has been accepted, The authors also give the decomposion temperature of C-S-H.
Reviewer 2 Report
The manuscript is now improved and I find it suitable for publication.